# Development of a MEL Cell-Derived Allograft Mouse Model for Cancer Research

**DOI:** 10.3390/cancers11111707

**Published:** 2019-11-01

**Authors:** Min Young Kim, Sungwoo Choi, Seol Eui Lee, Ji Sook Kim, Seung Han Son, Young Soo Lim, Bang-Jin Kim, Buom-Yong Ryu, Vladimir N. Uversky, Young Jin Lee, Chul Geun Kim

**Affiliations:** 1Department of Life Science, College of Natural Sciences, Hanyang University, Seoul 04763, Koreacswya@naver.com (S.C.); se2443@naver.com (S.E.L.); zojic93@hanmail.net (J.S.K.); imsangok12@naver.com (S.H.S.); dladudtn0@naver.com (Y.S.L.); 2Department of Clinical Pathology, Hanyang University Seoul Hospital, Seoul 04763, Korea; 3Department of Animal Science & Technology, Chung-Ang University, Ansung, Gyeonggi-do 17546, Korea; pado3607@daum.net (B.-J.K.); byryu@cau.ac.kr (B.-Y.R.); 4Department of Molecular Medicine, USF Health Byrd Alzheimer’s Research Institute, Morsani College of Medicine, University of South Florida, Tampa, FL 33612, USA; vuversky@health.usf.edu; 5Institute for Biological Instrumentation of the Russian Academy of Sciences, Federal Research Center “Pushchino Scientific Center for Biological Research of the Russian Academy of Sciences”, Pushchino, Moscow Region 142290, Russia; 6Institute of Pharmaceutical Science and Technology, Department of Pharmacy, Hanyang University, Ansan, Gyeonggi-do 15588, Korea; 7Research Institute for Natural Sciences, Hanyang University, Seoul 04763, Korea

**Keywords:** circulating tumor cells, erythroleukemia, allograft, liquid biopsy, cancer treatment

## Abstract

Murine erythroleukemia (MEL) cells are often employed as a model to dissect mechanisms of erythropoiesis and erythroleukemia in vitro. Here, an allograft model using MEL cells resulting in splenomegaly was established to develop a diagnostic model for isolation/quantification of metastatic cells, anti-cancer drug screening, and evaluation of the tumorigenic or metastatic potentials of molecules in vivo. In this animal model, circulating MEL cells from the blood stream were successfully isolated and quantified with an additional in vitro cultivation step. In terms of the molecular-pathological analysis, we were able to successfully evaluate the functional discrimination between methyl-CpG-binding domain 2 (Mbd2) and p66α in erythroid differentiation, and tumorigenic potential in spleen and blood stream of allograft model mice. In addition, we found that the number of circulating MEL cells in anti-cancer drug-treated mice was dose-dependently decreased. Our data demonstrate that the newly established allograft model is useful to dissect erythroleukemia pathologies and non-invasively provides valuable means for isolation of metastatic cells, screening of anti-cancer drugs, and evaluation of the tumorigenic potentials.

## 1. Introduction

Erythroleukemia is a subtype of acute myeloid leukemia (AML), accounting for 3~5% of AML cases. It is a neoplastic proliferation of erythroid and myeloid precursors of bone marrow hematopoietic stem cells [1]. It represents a phenotypically distinct form of AML characterized by unfavorable risk karyotype and disease features [2]. After a century of its descriptive history, many diagnostic, prognostic, and therapeutic implications related to this unique leukemia form remain uncertain. The rarity of the disease and the simultaneous involvement of associated myeloid compartment have complicated the in vitro studies of human erythroleukemia cell lines. In the 2016 World Health Organization (WHO) classification of myeloid neoplasms, erythroleukemia was merged with myelodysplastic syndrome, and pure erythroid leukemia (PEL) became the only type of acute erythroid leukemia [3]. Pure erythroid leukemia (PEL) is a rare and aggressive form of acute leukemia with poorly characterized biology [4]. Although data from pre-clinical studies using murine and cell line erythroleukemia models demonstrate that genetic changes, epigenetic alterations, and microRNA dysregulation are important for erythroleukemia pathogenesis, therapeutic strategies related to this unique leukemia form remain uncertain [2]. Murine erythroleukemia (MEL) cells are erythroid progenitors derived from the Friend virus-infected spleen [5,6], and their development is arrested at the proerythroblast stage. They can be maintained in vitro as proerythroblasts indefinitely, or until terminal differentiation signals are given by exposure to certain chemicals. Therefore, this cell line is one of the established cellular models of erythroid differentiation and the source of proerythroblasts. Although MEL cells are also commonly used as a model to dissect mechanisms of erythropoiesis and erythroleukemia in vitro, their tumorigenicity with homing properties in allograft mice is not used for cancer research as of yet.

In this study, we show that our established allograft model could offer analytic methods to quantify rarely circulating MEL cells in the blood stream and to validate roles of candidate molecules involved in erythropoiesis and tumorigenicity in vivo. Furthermore, our findings can point to novel strategies for anti-metastatic therapeutics and anti-cancer drug screening.

## 2. Results

### 2.1. Establishment of an Allograft Model and Evaluation of the Methyl-CpG-Binding Domain 2 (MBD2) and p66α Roles in Normal Erythroid Differentiation and Their Tumorigenic Potential in This Model

Mock (transfusion with phosphate buffered saline, PBS) as a control or green fluorescent protein (GFP)-expressing MEL cells (1 × 10^7^ cells/head) were transfused via tail vein injection into the athymic nude mice (BALB/c nu/nu) for establishing a MEL cell allograft model (Figure 1A). With no differences in total body and liver weight (Figure 1B), splenomegaly and splenic erythroleukemia (Figure 1C) were prominent at 14 days after transfusion (6 of 6 mice were affected, Figure 1D). In contrast, although most transfused GFP-MEL cells also had liver lesions (Figure 1E), they were not spread widely.

We found earlier that the Mi-2/nucleosome remodeling deacetylase (NuRD) chromatin remodeling complex (CRC) potentiates erythroid differentiation of proerythroblasts by regulating functions of the CP2c complex [7]. CP2c (also known as TFCP2, CP2, α-CP2, LSF, and LBP-1c) is a ubiquitously expressed transcription factor [8,9,10], exerting a critical role in globin expression and erythropoiesis [11,12,13,14]. The integrated Mi-2/NuRD CRC includes one chromodomain-helicase-DNA-binding protein, CHD (either CHD3 or 4), one histone deacetylase, HDAC (HDAC1 or 2), two deleted in oral cancer 1 (DOC1, also known as cyclin-dependent kinase 2-associated protein 1), three metastasis-associated, MTA (MTA1, 2, and 3), six nucleosome-remodeling factor subunit RBAP46 or RBAP48, two transcriptional repressor p66 (p66α or β), and MBD (MBD2 or 3) molecules [15]. The proper CRC assembly is critically mediated by the MBD2-p66α interaction [16,17]. Both Mbd2 and Mbd3 expression is down-regulated during differentiation of MEL cells in vitro and in normal erythropoiesis in mouse bone marrow, and Mbd2, but not Mbd3, down-regulation is crucial for erythropoiesis [7]. On the other hand, arbitral modulation of Mbd2 expression, but not those of Mbd3 or p66α, or inhibition of Mbd2-p66α interaction by the p66αΔ1 peptide induced both α- and β-globin expression and functional hemoglobin synthesis (about 25% of the normal differentiated MEL cells) by benzidine staining at the undifferentiated state [7], suggesting that MBD2-free NuRD functions as transcriptional coactivator for proper erythroid differentiation, while disruption of MBD2–NuRD by dissociation of the NuRD integrator p66α, does not induce functional hemoglobin synthesis at the undifferentiated state.

Here, we show that MEL cells with Mbd2 knock down (KD) or Mbd2/3 double knock down (DKD) by RNA interference significantly increased hemoglobin synthesis compared to that of wild-type (WT) or p66α KD cells, yet showing no effect on induced cells (Figure 2A). Short hairpin RNA (shRNA)-mediated p66α knockdown reduced the cell proliferation rate by the delayed G2/M-phase, arresting cells at G0/G1 phase (Figure 2B,C), suggesting that MBD2–NuRD is important for the proper proliferation of MEL cells, while MBD2-free NuRD induces spontaneous differentiation of MEL cells.

To confirm these phenomena in vivo using our established allograft model, the WT, Mbd2 KD, or Mbd DKD, as well as p66α KD MEL cells, were transfused into mice followed by the analysis at 14 days after transfusion. Splenomegaly was also not significant in mice transfused with Mbd2 KD, Mbd DKD, and p66α KD MEL cells (Figure 2D), with no differences in body and liver weight compared to the WT MEL cell-transfused mice (Figure 2E). In addition, splenic erythroleukemia was significantly reduced in these groups compared to the WT MEL cell-transfused mice (Figure 2F). Proliferating MEL cells were recovered by culturing cells collected from the circulating blood and spleen of both KD groups, showing ~20% of the WT MEL cell levels in spleen and no MEL cells in the blood circulation (Figure 2G), indicating that Mbd2 KD greatly reduced erythroleukemic potential of MEL cells. However, mice transfused with p66α KD MEL cells showed neither splenomegaly nor splenic erythroleukemia (Figure 2D,F). Importantly, although erythroid differentiation and splenomegaly were not observed in mice transfused with p66α KD MEL cells, proliferating MEL cells were recovered by culturing cells collected from the blood and spleen, showing ~50% of the WT MEL cell levels in spleen and in the blood circulation (Figure 2G). These observations suggest that although p66α KD MEL cells successfully homed to the spleen, they proliferated less efficiently compared to the WT MEL cells in the spleen, showing no splenomegaly. These effects were correlated with the aforementioned results of the in vitro cell cycle analyses (Figure 2B,C).

### 2.2. Erythroleukemia Cell-Transfused Mice as a Diagnostic Model to Evaluate Tumorigenicity Level and Metastatic Potentials in Cancer Researches

To test whether erythroleukemia cells in the blood stream could be employed as an indicator of the tumorigenicity level, GFP-expressing MEL or human erythroid leukemia (HEL) cells (each 1 × 10^7^ cells/head) were transfused via tail vein injection into the athymic nude mice (BALB/c nu/nu) for establishing MEL or HEL cell allograft or xenograft model, followed by collecting blood samples using eye bleeding at every 2 days after transfusion (Figure 3A). The number of GFP-expressing MEL or HEL cells at each stage was analyzed using Arthur™ fluorescence cell counter, and GFP-positive MEL or HEL cells were firstly detected at 2 days, then maintained stable cell numbers for 19 (MEL) or 14 (HEL) days after transfusion (Figure 3B,C).

In light of the characteristics of solid cancers that could be detached and metastasized to other organs from original sites via peripheral blood stream, GFP-expressing MEL or HEL cells (each 1 × 10^7^ cells/head) with Matrigel were transfused via subcutaneous injection into the athymic nude mice (BALB/c nu/nu) to induce solid cancer masses for evaluating applicability of the developed model in the diagnosis of tumorigenicity and metastatic potentials. Blood samples were collected from subcutaneously transfused mice using eye bleeding at every 2 days after transfusion (Figure 4A), and the number of GFP-expressing MEL or HEL cells at each stage was analyzed using Arthur™ fluorescence cell counter (Figure 4B). GFP-positive MEL or HEL cells were firstly detected at 6 days after transfusion, and then maintained thereafter. It is noted that circulating blood cell samples collected at 6 days after transfusion were not detected until in vitro primary cultivation, because of the low numbers of GFP-positive MEL or HEL cells (Figure 4C). In addition, the subcutaneously transfused mice with GFP-expressing MEL cells were sacrificed 17 days after transfusion, and splenomegaly was observed in MEL cell-transfused mice (Figure 4D). Meanwhile, splenomegaly was also prominent in MEL cell-transfused mice with tail vein or subcutaneous injection, suggesting the MEL allograft model is a useful model for tumorigenicity evaluation and analysis of the metastatic potentials.

### 2.3. The Allograft Model for Anti-Cancer Drug Screening and Validation

Since the extent of splenomegaly significantly correlates with the number of circulating MEL cells in the MEL cell allograft models (Figure 3 and Figure 4), our established allograft model can be used for anti-tumor drug screening, and it is expected that anti-cancer drugs can reduce both the extent of splenomegaly and the number of circulating MEL cells. GFP-expressing MEL cells were transfused into nude mice, and 20 mg/kg 5-Fluorouracil (5-FU) or 300 mg/kg *N*,*N*′-hexamethylene bisacetamide (HMBA) were injected at every 2 days for 10 days. The GFP-expressing MEL cells were collected from the circulating blood at day 8 after transfusion (day 6 after drug injection) and cultured for an additional 4 days with the analysis of cell number by fluorescence cell counter. The number of circulating MEL cells in 5-FU- or HMBA-treated mice was decreased compared to the number of circulating MEL cells in PBS-treated mice (Figure 5A). Furthermore, this analysis showed reduced splenomegaly in 5-FU- or HMBA-treated mice at 18 days after transfusion (Figure 5B). In our previous studies, 2-amino-*N*-(2,3-dihydro-benzo[1,4]dioxin-2-ylmethyl)-acetamide (ABA) was shown to have a role as an inhibitor of MBD2-p66α interaction, resulting in the inhibition of cancer metastasis, also resulting in the cell death of myeloid lineage cells [18]. Therefore, we employed this chemical for validating its effects on cancer treatment and for the analysis of the applicability of the established allograft model for anti-cancer drug screening. GFP-expressing MEL cells were transfused into nude mice, and different doses of ABA (12.2, 24.4, or 48.8 μg/kg) were injected at every 2 days for 10 days. The GFP-expressing MEL cells were collected, and the analyzed cell number and degree of splenomegaly are shown in Figure 4A,B, respectively. It is noted that treatment with ABA dose-dependently reduced both the number of circulating MEL cells (Figure 5C) and the extent of splenomegaly (Figure 5D), showing significant correlation among them. Therefore, our data strongly suggest that the established MEL cell allograft model could be used for screening anti-tumor drug candidates for future clinical applications. Furthermore, the developed method in this study of using fluorescence cell counter with cultivation provides a useful means for analysis of the in vivo efficacy of the anti-tumor drugs by counting circulating MEL cells that present in small numbers in the blood.

## 3. Discussion

Genetically engineered mouse models (GEMMs), cell line-derived xenograft (CDX) or patient-derived xenograft (PDX) models, and humanized mouse models (HMMs) have their own benefits and are commonly employed in cancer research [19,20,21,22,23,24,25,26,27,28,29,30]. Nevertheless, several issues in the development of animal models for cancer research have been encountered, such as different susceptibility of various mouse and rat strains to cancers, genetic diversions and technical variations in the animals, and polygenic predisposition in different strains controlled by multiple genetic loci [31,32,33,34,35,36,37,38]. Based on these and other limitations, currently used animal models might result in an inaccurate recapitulation of tumor development, leading to discrepancies between the outputs of preclinical research and clinical trials for development of cancer treatment [39,40,41]. Therefore, improvement of the relevance of animal models for the examination of human pathologies is urgently needed. As an example, in hepatocarcinogenesis studies, different susceptibility of various mouse and rat strains to liver cancer was shown to be dependent on variability of the capacity of initiated cells, modifier genes, and genetic loci controlling hepatocarcinogen resistance [42,43,44], and the best-fit mouse and rat models of hepatocarcinogenesis have been identified by comparative functional genetic studies [45,46,47]. Additionally, bone-defective animal models of the relevant species have been commonly used for studying bone regeneration and repair mechanisms, and these traditional bone regeneration techniques are applied to disease modeling with humanized bone or bone marrow transplantation in immunocompromised mice [33,48,49]. Here, we established the MEL cell-derived allograft model by intravenous (tail vein) or subcutaneous injection in immunocompromised BALB/c mice to study the pathogenesis of erythroleukemia. Although we did not use several mouse strains for validation of polygenic predisposition, and although this is not a xenograft model, our allograft mouse model is valuable for (1) providing fundamental knowledge for establishing the xenograft animal model for erythroleukemia, (2) developing detection methods of circulating tumor cells (CTCs) from the blood stream, (3) dissecting metastasis, and (4) screening or validating anti-cancer drugs.

Immunodeficient or syngeneic mice with transplanted cancer cells have been widely used as promising animal models in the evaluation of metastatic potentials [50,51,52,53,54,55]. For example, B16F10 cells (a murine melanoma cell line) have been injected subcutaneously or intravenously into syngeneic or immunocompromised mice for studying the mechanism of tumor formation and lung metastasis [56,57]. In our study, MEL cells derived from the Friend virus-infected spleen [5,6] were subcutaneously injected into the mice, resulting in a metastasizing tumor in the spleen (splenomegaly) of the animals. According to our findings and previous reports, the developed MEL cell-derived animal model could be applied for evaluating metastatic potentials through analysis of the circulating cells in the blood stream. In line with the previous reports showing that xenograft models represent more natural tumor conditions compared to in vitro models [58], and allow developing anti-tumor drugs, biomarkers, and novel therapeutic methods [59], we established here allograft and xenograft models of erythroleukemia using MEL and HEL cells followed by isolation, cultivation, and quantification of circulating MEL and HEL cells and analysis of the affected organs. The MEL cell allograft showed higher tendency of splenomegaly when compared to the HEL cell xenograft, although both models showed similar mode and extent of CTCs. These observations suggest that the MEL cell allograft model is superior to the HEL cell xenograft model for the evaluation of anti-tumor drugs or biomarkers, and for the development of novel therapeutic methods. Our observations are consistent with the previous findings that the tumorigenicity of MEL cells depend on the site of tumor growth, and that MEL cells multiply to a limited extent in the liver, with marked differentiation to the orthochromatic erythroblast stage [60]. Therefore, we hypothesize here that MEL cells have a remarkable potential to home and vigorously proliferate in the spleen, leading to splenomegaly. Furthermore, the proliferated MEL cells could also be liberated into the blood stream.

Regarding employment of MEL cells as a cellular model of erythropoiesis [7,8,9,10,11,12,13,14,15,16,17], we successfully confirmed that Mbd2 and p66α have different roles in erythroid differentiation and tumorigenicity in vivo (Figure 2). Therefore, our MEL cell-transfused allograft model provides a valuable tool to discriminate between erythroid differentiation and tumorigenic potential of erythroleukemia by examining splenomegaly and circulating cells in the blood. Also, these data suggest that our established allograft model could be useful to analyze and validate in vitro data, and also, can be expanded to other practices.

To check the applicability of this model for validation of the effects of anti-tumor drugs, 5-FU, HMBA, or ABA was injected into the MEL-transfused mice. We observed that the number of the isolated MEL cells in the blood stream of each drug-treated mouse (after the additional in vitro cultivation) was decreased dose-dependently compared to the control. Based on the observations reported here, the usefulness of our allograft model is two-fold (Figure 6). First, relatively constant and high numbers of MEL cells (circulating tumor cells, CTCs) were observed in the blood from day 6 or day 10 after transfusion of MEL cells into tail vein or transplantation of cells by subcutaneous injection to the flank, respectively. Since splenomegaly was prominent in allograft mice during our experimental period, it is hypothesized that some CTCs were constantly replenished from the spleen as surplus cells were degraded as normal erythrocytes do. Therefore, this kinetics of CTCs provides our allograft model applicability for testing anti-cancer drugs targeting erythroleukemia or pan-cancers, otherwise it is hard to detect rare CTCs in other models. Second, splenomegaly formation by homing MEL cells into the spleen is another indicator for testing efficiency of anti-cancer drugs targeting erythroleukemia or pan-cancers. Based on these results, we suggest that our allograft model based on the use of MEL cells, combined with isolation, cultivation, and quantification of circulating tumor cells, represents a reliable and useful tool for the development of anti-tumor drugs, preclinical diagnosis, and analysis of biomarkers for cancer treatment.

Recently, several challenges continue to hamper the efficiency of the CTC-based diagnosis and models, such as development of novel approaches for detection and quick isolation/enrichment of rare CTCs in patient blood streams. Therefore, the interest is increasing in the development of new technologies involved in management of CTCs. For example, although we employed in this study Arthur™ fluorescence cell counter for analyzing GFP-positive circulating MEL cells, we could have analyzed non-marked CTCs using recently developed techniques, such as a slanted weir microfluidic device [61]. We recognize that additional studies are needed to further improve our allograft model that can be utilized to easily isolate metastatic cells, target metastatic progression, evaluate tumorigenic potentials of target genes, screen drugs, and develop patient-specific treatments.

## 4. Materials and Methods

### 4.1. Cell Culture

The MEL DS19 or HEL cell line was obtained from Dr. Mark Groudine, Fred Hutchinson Cancer Research Center, USA or Dr. YoungYiul Lee, Hanyang University Medical School, Republic of Korea, respectively. They were cultured in Dulbecco’s modified Eagle’s medium (DMEM, Hyclone SH30243.01) or Roswell Park Memorial Institute 1640 (RPMI-1640) medium X1 (Hyclone SH30027.01), supplemented with 10% fetal bovine serum (FBS, Hyclone, SH30084.03) and 1% penicillin/streptomycin, respectively. All cells were grown at 37 °C under an atmosphere containing 5% CO_2_. For stable cell line establishment, the constructs were transfected into MEL/HEL cells using Effectene reagent (Qiagen, 301425, Hilden, Germany). An Olympus IX71 microscope was used to measure the green fluorescence of GFP-tagged MEL/HEL cells when excited at 485 nm and emitted at 530 nm. Erythroid terminal differentiation experiments and benzidine staining have been shown previously [7].

### 4.2. Plasmid Constructs

The pLV-TH plasmid was used to establish GFP-tagged MEL or HEL cell lines. The pEF1α-3XFB-p66α full length, pEF1α-3XFB-p66α Δ1, pSuper-Puro-shp66α, pCMV-Tag1-Myc-Mbd2 (a generous gift of Dr. Gerd P. Pfeifer, Beckman Research Institute of City of Hope, USA), and pSuper-Puro-shMbd3 plasmids have been described previously [7]. The previously constructed pQCXIP-shMbd2 vectors (a generous gift of Dr. James Hagman, National Jewish Health, USA) were used for the expression of *MBD2*-targeting shRNA.

### 4.3. Animal Studies

For visualization of the number of blood circulating leukemia cells, 6-week-old BALB/c nude mice (Nara Biotec, Seoul, Korea) were randomly divided into 3 mice per injection group. Approximately 100 μL PBS of single cell suspension (1 × 10^7^ cells) of GFP-tagged MEL/HEL cell lines with Matrigel at a ratio of 1:2 were injected via the tail vein or by subcutaneous injection. For cell proliferation assay, peripheral blood cells were obtained by eye bleeding from retro-orbital plexus blood (100 μL) once every 2 days. Peripheral blood cells and single cell suspensions derived from spleen tissue slices (0.01 g) were cultured for 8 days in DMEM or RPMI-1640 supplemented with penicillin/streptomycin, in the presence of 10% FBS. Cell numbers were counted by an image base cytometer (NanoEnTek Arthur™, Seoul, Korea) using a quick count program or a green fluorescence count program. Mice were transfused with GFP-expressing MEL cells with 5-FU (Sigma-Aldrich, Saint Louis, MO, USA; 20 mg/kg), HMBA (Sigma-Aldrich; 300 mg/kg), or ABA (Fluorochem, Derbyshire, UK; 12.2, 24.4, and 48.8 μg/kg) in PBS by tail vein injection and at every 2 days after transfusion for 10 days. The GFP-expressing MEL cells were collected from circulating blood at 8 days after the transfusion, and the number of cells was measured at that time by fluorescence cell counter (NanoEnTek Arthur^TM^). In addition, when the contents of GFP-expressing cells were not high enough to be detected directly by the fluorescence cell counter at early days after transfusion, collected cells were cultured and the cell number was analyzed at every couple of days for 8 days. At 14 days after transfusion, mice were sacrificed, and the presence of splenomegaly was analyzed.

For histological analysis, major organs of mice were immediately fixed in 10% formalin neutral buffer solution, dehydrated in a graded series of ethanol, treated with xylene, paraffin embedded, and cut into 4 μm sections. Hematoxylin and Eosin (H/E)-stained tissue sections were analyzed and photographed with a light microscope, and the representative histological images were recorded at ×400 magnification. All animal procedures were approved by the Animal Care and Use Committee and the Institutional Review Board of Chung-Ang University (Approval No., 2015-00022; Approval Date, 13 May 2015).

### 4.4. Cell Cycle Analysis

WT and p66α KD cells were washed and suspended with pre-chilled PBS, and then fixed with pre-chilled 70% ethanol at −20 °C. Cells were then incubated with 20 μg/mL RNase A at 37 °C for 30 min and stained with 50 μg/mL propodeum iodide (PI) at room temperature for 20 min. Samples were immediately analyzed by flow cytometry with a FACSCalibur flow cytometer (BD Biosciences, San Jose, CA, USA) with FlowJo software (BD Biosciences).

### 4.5. Quantification and Statistical Analysis

Data are presented as the mean ± standard error. The sample size for each experiment, *n*, is included in the results section and the associated figure legend. Everywhere in the text the difference between two subsets of data is considered statistically significant if the one tailed Student’s *t*-test gives a significance level (*p*-value) less than 0.05. Multiple comparisons were performed using ANOVA, where a Scheffe post-test was performed in some cases, or a Kruskal–Wallis test. Statistical analyses were performed using IBM SPSS Statistics 23 (IBM, Armonk, NY, USA).

## 5. Conclusions

Although MEL cells are commonly used as a model to dissect mechanisms of erythropoiesis and erythroleukemia in vitro, their tumorigenicity with homing properties in allograft mice is not used for cancer research as of yet. In this study, we established a MEL cell allograft model and evaluated it as a diagnostic model for isolation/quantification of circulating MEL cells, anti-cancer drug screening, and validation of the tumorigenic potentials of molecules in vivo. We show here that a MEL cell allograft model is useful to dissect erythroleukemia pathologies, and provides valuable means for screening of anti-cancer drugs and evaluation of tumorigenic potentials.

## Figures and Tables

**Figure 1 cancers-11-01707-f001:**
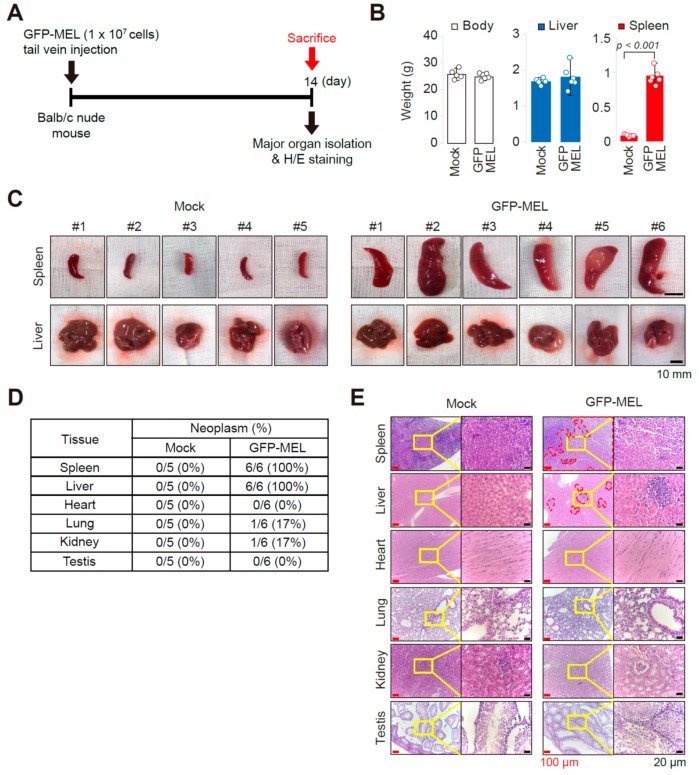
Establishment of an allograft mouse model using GFP-MEL cells. (**A**) Scheme of organ collection at 14 days after tail vein injection of GFP-expressing MEL cells. (**B**) Evaluating splenomegaly by quantification of body, spleen, and liver weight of each group of mice. Mock (transfused with phosphate buffered saline), *n* = 5; GFP-MEL, *n* = 6. *p* < 0.001 by one-tailed *t*-test to corresponding mock-treated group. (**C**) Photographs of spleens and livers dissected out from the mock (left; *n* = 5) and GFP-MEL cells transfused mice (right; *n* = 6) at 14 days after systemic injection. (**D**) Neoplasm (erythroleukemia) incidences in the major organs of mice. (**E**) Photographs showing representative H/E-stained tissue sections for the major organs, with highly magnified images of yellow square areas. Regions with red dot spots indicate lesions with transfused GFP-MEL cells in the liver and spleen.

**Figure 2 cancers-11-01707-f002:**
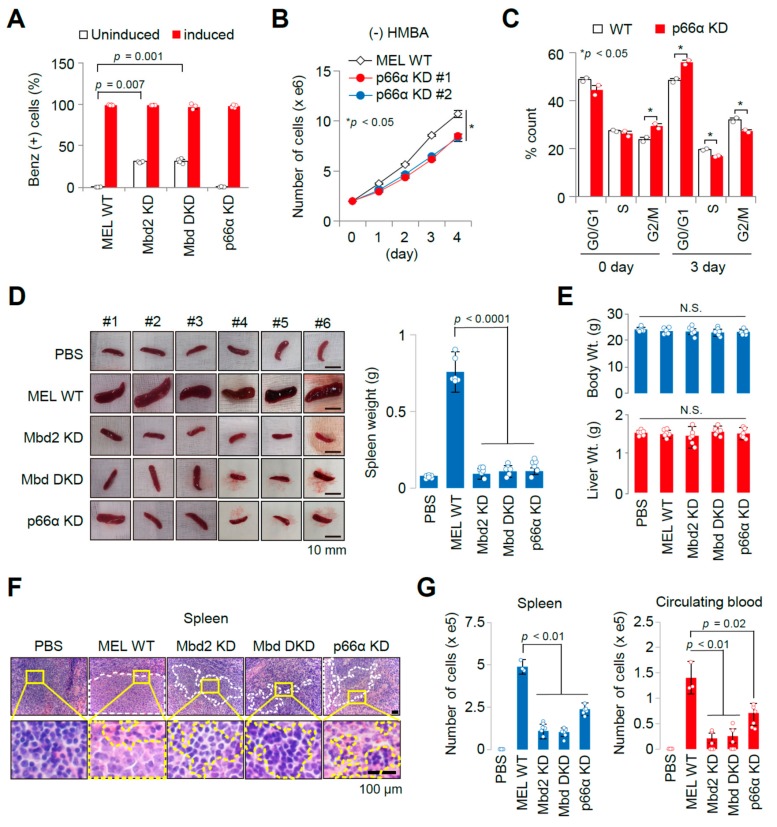
Evaluating Mbd2 and p66α roles in tumorigenic potential in vivo by established allograft model. (**A**) Functional hemoglobin synthesis analysis in the wild-type (WT) MEL cell or in MEL cells with various modulations of the Mi-2/NuRD components (Mbd2 KD, Mbd DKD, p66α KD) by benzidine staining. Fractions of benzidine stain-positive cells were measured at undifferentiated (d0) or differentiated (d3) state by HMBA treatment in vitro. *n* = 4. Significance test among each cell line relative to the WT cells was done using univariate analysis of variance (ANOVA). Cell proliferation (**B**) and cell cycle distribution (**C**) analysis of WT and p66α KD MEL cell lines (*n* = 2). Reduction of cell proliferation potential in p66α KD MEL cells is due to cell cycle arrest at G2/M phase. *; *p* < 0.05, by ANOVA (**B**) or one-tailed *t*-test (**C**). (**D**) Photographs (left) and quantification of spleens dissected out from each group at 14 days after systemic injection mice (**C**). *n* = 6 or 3/group. Significance was tested by ANOVA. (**E**) Measurement of body and liver weight of mice groups at 14 days after transfusion of WT or various MEL cell lines (mock, Mbd2 KD, Mbd DKD, and p66α KD). (**F**) Photographs showing representative H/E-stained tissue sections of spleen derived from each cell line-transfused mouse, with highly magnified images of yellow square areas. Regions with white and yellow dot spots indicate lesions with transfused GFP-MEL cells in liver and spleen. It is noted that there are no apparent MEL cells in the spleens, except those of WT MEL cell-transfused mice. (**G**) Estimation of the number of MEL cells harbored in the spleen (left) and circulating in the blood of allograft mice. *n* = 6/group. Significance was tested by ANOVA.

**Figure 3 cancers-11-01707-f003:**
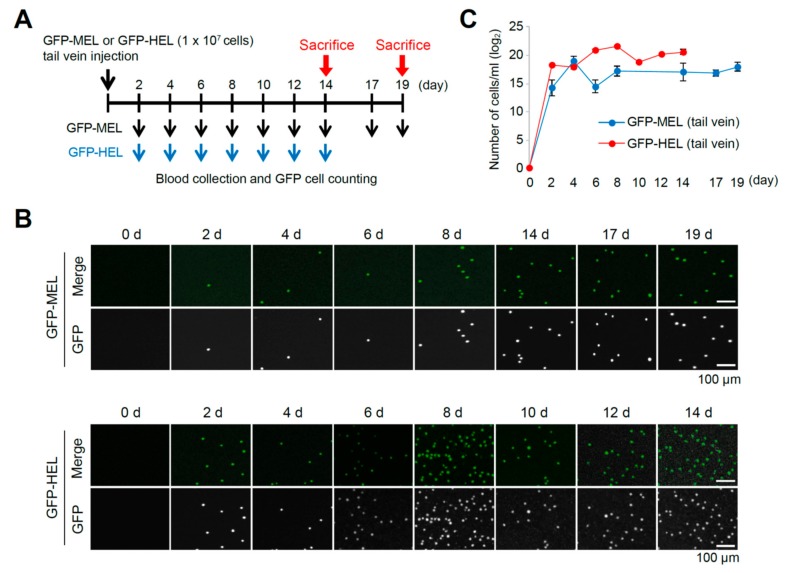
Analysis of circulating MEL or HEL cells by tail vein injection to validate applicability for tumorigenic level. (**A**) Scheme of blood collection by eye bleeding for 14 or 19 days after GFP-expressing MEL or HEL cell transfusion, respectively. (**B**) Representative Arthur™ (NanoEnTek Arthur ^TM^, Seoul, Korea) histograms show the green fluorescence intensities and counts for the GFP-expressing MEL or HEL cell. (**C**) Quantification of collected circulating GFP-positive MEL or HEL cells at every 2 days after transfusion for 14 (HEL cells) or 19 days (MEL cells; additional 2-day cultivation after euthanization) that showed stable cell numbers in blood stream at 48 h after transfusion.

**Figure 4 cancers-11-01707-f004:**
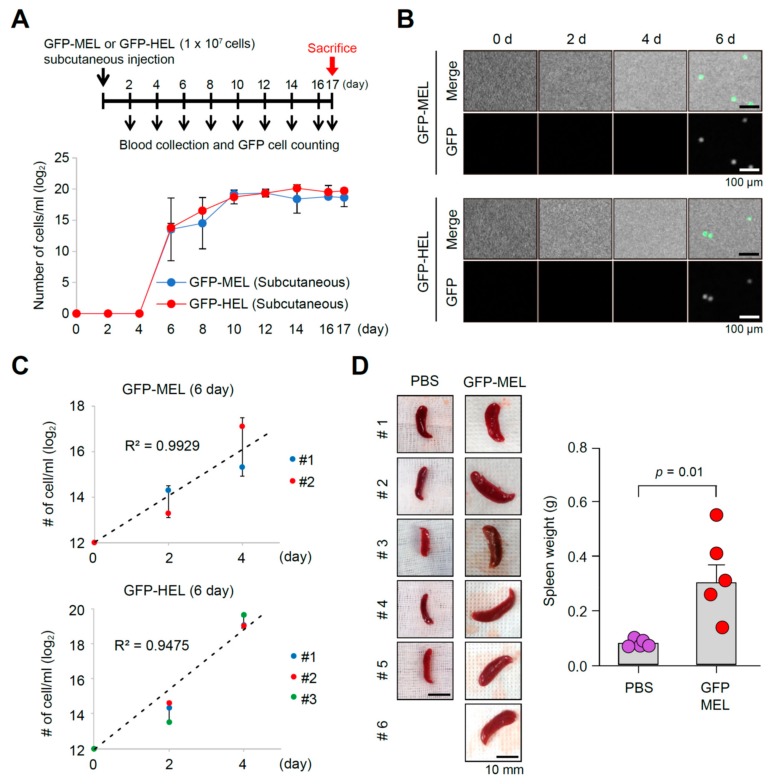
Analysis of erythroleukemia cell-transfused mice by subcutaneous injection as diagnostic models for validating tumorigenicity and metastatic potential. (**A**) Scheme of blood collection by eye bleeding for 17 days after GFP-expressing MEL or HEL cell transfusion using subcutaneous injection. Quantification of collected circulating GFP-positive MEL or HEL cells at every 2 days after transfusion for 17 days that showed stable cell numbers in blood stream at 144 h after transfusion. (**B**) Representative photographs showing the green fluorescence intensities and counts (panel A) for the GFP-expressing MEL or HEL cells in the blood sample by Arthur™ (NanoEnTek Arthur TM, Seoul, Korea) histograms at every 2 days during 6 days. (**C**) Quantification curve of GFP-positive MEL at every 2 days during additional 4-day cultivation, the circulating blood sample collected at 6 days after transfusion. (**D**) Photographs of spleens dissected out from the mock (top; *n* = 5) and GFP-expressing MEL (middle; *n* = 6) cell-transfused mice (right; *n* = 6) at 17 d after systemic injection. Splenomegaly was only observed in GFP-expressing MEL cell-transfused mice. Significance of each combination was tested by Student’s *t*-test.

**Figure 5 cancers-11-01707-f005:**
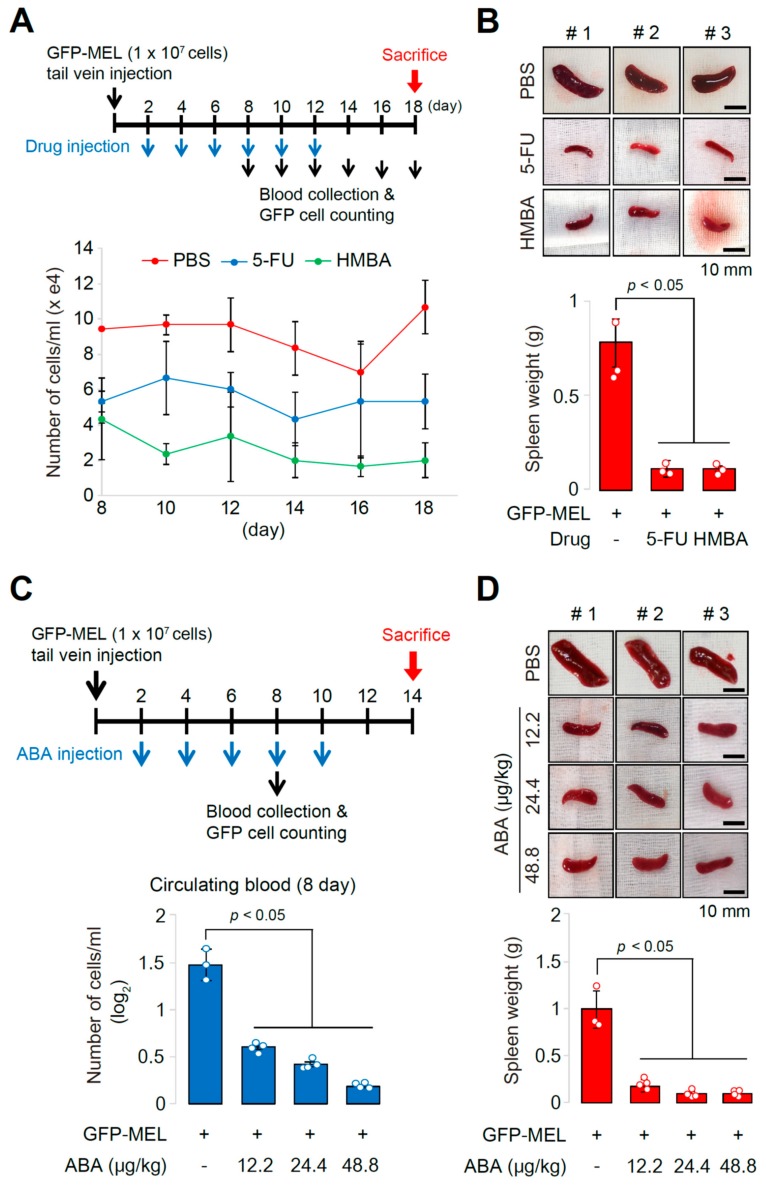
Applicability of the established allograft model and circulating blood analysis method for screening and validation of anti-cancer drugs. (**A**) Timeline of animal experiments for allograft model (intravenous injection of GFP-expressing MEL cells), drug administration, and circulating blood analysis. Mice were transfused with GFP-expressing MEL cells, and 5-FU (20 mg/kg) or HMBA (300 mg/kg) was intravenously administrated at every 2 days after transfusion for 12 days. At 8 days after transfusion, blood samples from each group were collected at every 2 days for 10 days to measure cell numbers in blood stream of each group. Quantifications of GFP-positive MEL cells at each stage of blood collections are shown. Compared to the cell number of GFP-positive MEL cells with phosphate buffered saline (PBS), those with 5-FU or HMBA were shown to be decreased. (**B**) Photographs of spleens dissected out from the PBS-, 5-FU-, or HMBA-treated transfused mice (*n* = 3) at 18 days after systemic injection. Splenomegaly was only observed in PBS-treated transfused mice with comparison of the spleen weight by ANOVA. (**C**) 2-amino-N-(2,3-dihydro-benzo[1,4]dioxin-2-ylmethyl)-acetamide (ABA with different concentrations (12.2, 24.4, or 48.8 μg/kg) was administrated to transfused mice at every 2 days for 10 days, and circulating MEL cell numbers collected from the blood at day 8 after transfusion (at day 6 after ABA administration), followed by additional cultivation for 4 days. The number of GFP-expressing MEL cells in the blood of transfused mice was decreased in the blood of ABA-administrated mice, showing a dose-dependent manner. (**D**) Photographs of spleens dissected out from the PBS, different concentrations of ABA-treated transfused mice at 14 days after systemic injection. Splenomegaly was only observed in PBS-treated transfused mice with comparison of the spleen weight by ANOVA.

**Figure 6 cancers-11-01707-f006:**
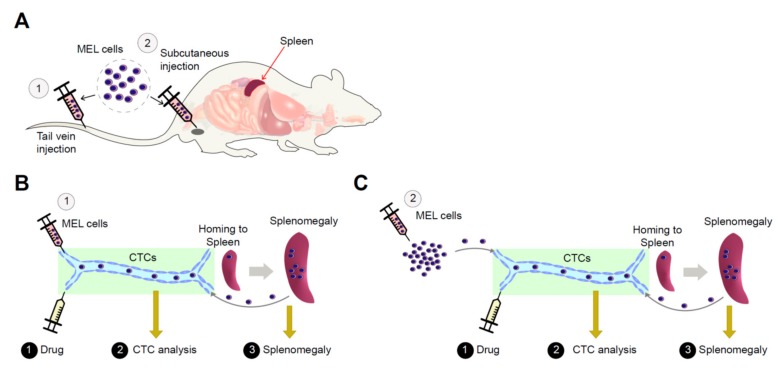
Schematic overview of established allograft model and its applicability for cancer researches. (**A**) Development of MEL cell-derived allograft model mice by ① tail vein or ② subcutaneous injection. Approaches to cancer detection, diagnosis and anti-cancer drug discovery by isolation and enrichment of MEL cells (CTC) with examination of splenomegaly in tail vein- (**B**) and subcutaneously-injected (**C**) MEL cell-derived allograft model.

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
