# Peer review of "Development of a MEL Cell-Derived Allograft Mouse Model for Cancer Research"

_cancers, 2019, doi:10.3390/cancers11111707_

Round 1

Reviewer 1 Report

This study has some issues regarding the choice of experimental system resulting in over-interpretation of the results.

The Introduction starts by talking about cancer as a deadly disease followed by introducting CTC of solid tumors and then switching to erythroleukemia. The text lacks focus and has a lot of general statements that are simply incorrect. MEL cells as a model for solid tumors in the first place is simply not valid. Blood cells can of course survive in blood so MEL cells have very little in common with CTC. The different transplant methods does not help. The authors call this an allograft model. I have difficulties agreeing that one can discuss immunogenicitiy and allografting in a model (Nude mice) without adaptive immune cells. If ordinary Balb/c mice were used then it was fine but now the model is not so impressive since even human cells can grow in Nude mice.

Taken together, the study would benefit from more focus on one topic, namely using the model to test therapies against erythroleukemia. In that case follow up using human cells would also bee needed. The data on CTC has too little bearing on the clinical counterpart to be acceptable for publication.

Author Response

Thank you for your kind comments. The manuscript has been revised to address the raised concerns. We rearranged and corrected phrases in Abstract, Introduction and Discussion. We paid special attention to the use of MEL cells as a model for solid tumor to analyze metastasis. We added corresponding statements to the Discussion section (highlighted blue).  Please see the attachment.

Reviewer 2 Report

Reviewer Comments to the Author:

In the present study, Min Young Kimet al. showed that the allograft and xenograft models could offer analytic methods to quantify rare circulating MEL cells in blood stream and to validate roles of candidate molecules involved in erythropoiesis and tumorigenicity in vivo. This study shows that these murine models could be used to develop strategies for anti-metastatic therapeutics and anti-cancer drug screening. 

Although erythroleukemia is very rare disease, this study adds new information to the field. The current study includes the following pointsthat need to be addressed for the improvement of the manuscript.

Major comments:

Introduction:

Lines 43:  “Cancer is an incurable disease and ”  This is not accurate description. Please revise this sentence.

The frequency of erythroleukemia should be noted in introduction.

Line 46: “Circulatorytumor cells (CTCs)” : CTC is generally “Circulating tumor cells”. This should be revised.

Line 56-57: “the number and proliferation rate of CTCs collected from blood stream have been correlated with the cancer mass, level of malignancy, and metastatic propensity”.   This sentence should be revised.  Especially, “cancer mass” and “level of malignancy” are unclear words and inappropriate terms.

Line 71: “PEL is a rare and aggressive form of acute leukemia with poorly characterized biology”. 

What is the PEL?  Is this sentence correct?

Methods:

Did the authors try to isolate and count directly instead of culture blood?

GFP-labeled CTC isolation and counting CTCs in hematological malignancy should be easier than solid tumor.

Author Response

We are thankful to this reviewer for careful reading of the manuscript and for kind comments. The manuscript has been revised to address all the indicated issues.

Introduction:

Lines 43:  “Cancer is an incurable disease and” This is not accurate description. Please revise this sentence.

This sentence has been removed to refine and focus on our findings.

The frequency of erythroleukemia should be noted in introduction.

The frequency of erythroleukemia has been noted in Introduction. (highlighted yellow)

Line 46: “Circulatory tumor cells (CTCs)”: CTC is generally “Circulating tumor cells”. This should be revised.

We have corrected “circulatory” to “circulating” (highlighted yellow)

Line 56-57: “the number and proliferation rate of CTCs collected from blood stream have been correlated with the cancer mass, level of malignancy, and metastatic propensity”.   This sentence should be revised.  Especially, “cancer mass” and “level of malignancy” are unclear words and inappropriate terms.

 This sentence has been removed to refine and focus on our findings.

Line 71: “PEL is a rare and aggressive form of acute leukemia with poorly characterized biology”. 

What is the PEL?  Is this sentence correct?

We have revised this sentence by adding the meaning of the abbreviation PEL, which is Pure erythroid leukemia (PEL) (highlighted yellow)

Methods:

Did the authors try to isolate and count directly instead of culture blood?

GFP-labeled CTC isolation and counting CTCs in hematological malignancy should be easier than solid tumor.

Methods for isolation and counting of CTCs have been described in section 4.3 Animal studies in Materials and Methods as follows, “the GFP-expressing MEL cells were collected from circulating blood at 8 days after transfusion and cultured with analysis of cell number at every couple of days for 8 days by fluorescence cell counter (NanoEnTek ArthurTM).”

Please see the attachment of our revised manuscript.

Round 2

Reviewer 2 Report

The authors did not address a comment from the reviewer.

Method for CTC detection:

"the GFP-expressing MEL cells were collected from circulating blood at 8 days after transfusion and cultured with analysis of cell number at every couple of days for 8 days by fluorescence cell counter (NanoEnTek Arthur)."

The reviewer want to know why the authors needed to culture the blood in vitro to detect CTCs.

Culturing CTCs may induce growth of CTC cell line and the authors may not be able to evaluate accurate CTC numbers.

Did the authors count CTC numbers just after blood drawn?

Author Response

Thank you for your kind comments and sorry for confusion.

Yes, the CTC numbers were measured just after blood was drawn at 8 days after the transfection. We are describing the count method in Materials & Methods (4.3 Animal studies).
